# Resistance Genes, Plasmids, Multilocus Sequence Typing (MLST), and Phenotypic Resistance of Non-Typhoidal *Salmonella* (NTS) Isolated from Slaughtered Chickens in Burkina Faso

**DOI:** 10.3390/antibiotics11060782

**Published:** 2022-06-08

**Authors:** Assèta Kagambèga, Elizabeth A. McMillan, Soutongnooma C. Bouda, Lari M. Hiott, Hazem Ramadan, Daniel K. Soro, Poonam Sharma, Sushim K. Gupta, Nicolas Barro, Charlene R. Jackson, Jonathan G. Frye

**Affiliations:** 1Bacterial Epidemiology and Antimicrobial Resistance Research Unit, U.S. National Poultry Research Center, Agricultural Research Service, United States Department of Agriculture, Athens, GA 30605, USA; elizabeth.mcmillan@usda.gov (E.A.M.); lari.hiott@usda.gov (L.M.H.); hazem.ramadan@usda.gov (H.R.); charlene.jackson@usda.gov (C.R.J.); 2Laboratoire de Biologie Moléculaire, D’épidémiologie et de Surveillance des Bactéries et Virus Transmissibles par les Aliments (LaBESTA)/Ecole Doctorale Sciences et Technologies (EDST)/Université Joseph KI-ZERBO, Ouagadougou 03 BP 7021, Burkina Faso; boudacaroline@gmail.com (S.C.B.); karna_daniel@yahoo.fr (D.K.S.); barronicolas@yahoo.fr (N.B.); 3Institute des Sciences, Ministère des Enseignement Supérieur, de la Recherche Scientifique et de L’innovation, Ouagadougou 03 BP 7021, Burkina Faso; 4Hygiene and Zoonoses Department, Faculty of Veterinary Medicine, Mansoura University, Mansoura 35516, Egypt; 5Institute of Biosecurity and Microbial Forensics, Department of Biochemistry and Molecular Biology, Oklahoma State University, Stillwater, OK 74078, USA; poonam.sharma@okstate.edu; 6Department of Biochemistry and Molecular Biology, Oklahoma State University, Stillwater, OK 74078, USA; sushim.gupta@okstate.edu

**Keywords:** *Salmonella*, genomics, antimicrobial resistance, chicken

## Abstract

The emergence of antimicrobial-resistant bacteria in developing countries increases risks to the health of both such countries’ residents and the global community due to international travel. It is consequently necessary to investigate antimicrobial-resistant pathogens in countries such as Burkina Faso, where surveillance data are not available. To study the epidemiology of antibiotic resistance in *Salmonella*, 102 *Salmonella* strains isolated from slaughtered chickens were subjected to whole-genome sequencing (WGS) to obtain information on antimicrobial resistance (AMR) genes and other genetic factors. Twenty-two different serotypes were identified using WGS, the most prevalent of which were Hato (28/102, 27.5%) and Derby (23/102, 22.5%). All strains analyzed possessed at least one and up to nine AMR genes, with the most prevalent being the non-functional *aac*(6′)-Iaa gene, followed by *aph*(6)-Id. Multi-drug resistance was found genotypically in 36.2% of the isolates for different classes of antibiotics, such as fosfomycin and β-lactams, among others. Plasmids were identified in 43.1% of isolates (44/102), and 25 plasmids were confirmed to carry AMR genes. The results show that chicken can be considered as a reservoir of antibiotic-resistant *Salmonella* strains. Due to the prevalence of these drug-resistant pathogens and the potential for foodborne illnesses, poultry processing and cooking should be performed with attention to prescribed safe handling methods to avoid cross-contamination with chicken products.

## 1. Introduction

Foodborne illness caused by *Salmonella* is a public health concern around the world [1]. *Salmonella* are zoonotic bacteria and show increasing resistance to antibiotics, especially those isolated from farm animals. Several researchers have shown that *Salmonella* circulate between animals, humans, and the environment [2,3,4]. Chickens constitute a reservoir of *Salmonella*, which can be part of the normal intestinal flora of these birds [5]. In Burkina Faso, chicken farming is a booming industry, and the consumer demand for chicken is increasing every year. In Ouagadougou (population ~3 million), the capital city of Burkina Faso, it is estimated that more than 80,000 chickens are slaughtered every day for consumption [6]. Chicken farming has become a good source of income for many Burkinabe who are carrying out this activity without adequate training on good farming hygienic practices. In this country, animal feed is produced in an unregulated manner without veterinary checks or microbiological quality analysis. In addition, there are feed producers who add antibiotics to their feed to prevent disease and the loss of flocks. These practices could select for antibiotic resistance in the bacteria of the chickens’ intestinal flora. This has greatly affected the emergence of multi-drug-resistant strains of *Salmonella*. The slaughter conditions of chickens constitute a critical point of contamination of carcasses, especially during evisceration [7]. After slaughter, the carcasses are exposed to ambient temperature for sale all day. These practices undoubtedly constitute the critical points of the contamination of humans by pathogenic bacteria from chickens [7,8]. Previous data have shown the presence of antibiotic-resistant *Salmonella* in Burkina Faso, but very little data exist on the genomic characteristics of these *Salmonella* due to a lack of equipment and methods in Burkina Faso [9,10]. Whole-genome sequencing (WGS) allows for the analysis of the *Salmonella* genome but requires expensive equipment and reagents, which are inaccessible to laboratories in Burkina Faso; therefore, for this study, we collaborated with scientists at the United States Department of Agriculture, Agricultural Research Service (USDA-ARS). Previously, we investigated the genomic characterization of *Salmonella* isolated from fish in Burkina Faso [11]. The present study aims to characterize *Salmonella* strains isolated from slaughtered chickens in Burkina Faso using WGS to better understand their molecular epidemiology.

## 2. Results

### 2.1. Serotypes of Isolates from Slaughtered Chickens

Twenty-two different serotypes were identified using WGS data. The most prominent serotype found was Hato with 28 (27.5%) isolates, followed by Derby with 23 (22.5%); Muenster with 7 (6.9%); and Typhimurium, Poona, Chester, and Kentucky all with 4 (3.9%). Other serotypes found include Alexanderplatz and Bredeney with three isolates (2.9%) each, and Rechovot, Telelkebir, and Tennessee with two (2.0%) each. Five isolates (4.9%) were named with their antigenic formula, and one isolate serotype could not be distinguished between Albany and/or Dusseldorf (Table 1 and Figure 1).

### 2.2. Antimicrobial Resistance Genes and Antibiotic Resistance Phenotypes Detected

Antibiotic resistance genes and phenotypic resistance profiles are shown in Table 1 and Figure 1. Phenotypic resistance to at least one antibiotic was observed among 33 strains (32.4%), and 27 (27/33, 81.8%) of those were multi-drug resistant (MDR). All isolates possessed at least the aminoglycoside resistance gene *aac*(6′)-Iaa. Other aminoglycoside resistance genes found included *aph*(3′)-Ia, *aac*(3)-Id, *aph*(6)-Id, *aph*(3′’)-Ib, *aad*A1, and *aad*A7.

Multiple resistance genes were found in 37 (36.3%) of the strains, with each strain possessing two to nine genes. Nineteen (19/28) *S.* Hato isolates possessed seven to nine resistance genes, including up to five aminoglycoside resistance genes, *sul*2 conferring resistance to sulfonamides, and *tet*(A) conferring resistance to tetracycline, and two contained *dfr*A14 conferring resistance to trimethoprim. Twenty-one of the *S*. Hato isolates possessed the *sul*2 gene, and nineteen had the *dfr*A gene. Twenty of these isolates showed phenotypic resistance to trimethoprim/sulfamethoxazole. Of the 19 isolates possessing the *tet*(A) gene, 18 showed phenotypic tetracycline resistance. However, in 15 of the 28 *S*. Hato isolates, at least one AMR gene was a partial sequence (Table 1).

Of the 23 *S*. Derby isolates, 7 contained multiple resistance genes. Five isolates had five resistance genes, and one isolate had six resistance genes, including multiple aminoglycoside genes, *sul*2, *tet*(A), and *fos*A7. Two *S*. Derby isolates had the *fos*A7 gene. The six *S*. Derby isolates with *tet*(A) were phenotypically resistant to tetracycline, with four also being resistant to Minocycline. 

All seven *S.* Muenster isolates had only one resistance gene, *aac*(6′)-Iaa, but no aminoglycoside resistance was seen. Of the four *S.* Typhimurium isolates, one possessed *bla*_TEM-1B_, conferring resistance to β-lactams, and *mph*(A), suggesting resistance to macrolides. However, no macrolides were tested on this panel. Interestingly, one *S*. Typhimurium isolate with only the *aac*(6′)-Iaa gene found by WGS showed resistance to tetracycline and trimethoprim/sulfamethoxazole. Of the four *S.* Kentucky isolates, three had nine resistance genes conferring resistance to aminoglycosides, sulfonamides, trimethoprim, tetracyclines, and quaternary ammonium compounds (*qac*E). The *qac*E gene was partially present in all *S*. Kentucky strains, which may suggest resistance to sanitizers such as benzalkonium chloride. The chromosomal mutations and MLST results of the strains are shown in Table 1. Twenty-eight different MLST types were identified. Multiple MLST types were identified among serotypes Derby, Hato, Poona, Telelkebir, and Typhimurium. Some isolates of the serotypes Alexanderplatz, Brancaster, Derby, Farmingdale, Hato, I 1,3,19:b:-, I 1,3,19:f,g:1,5, and Rechovot had unknown MLST types.

Among the isolates with multiple AR genes, strong positive correlations (r > 0.7) were found for the co-occurrence of several AR genes (Figure 2). Strong positive correlations were found for the co-occurrence of multiple aminoglycoside genes, including *aad*A1 with *aph*(3′)-Ia, *aph*(3″)Ib, or *aph*(6″)-Id. Strong correlations were also found for the co-occurrence of antibiotic resistance genes from different antibiotic classes, including *sul*2 with multiple aminoglycoside resistance genes, *tet*A, or *drf*A14 (Figure 2).

### 2.3. Replicon Types Detected

In this study, 43.1% (44/102) of *Salmonella* strains possessed at least one plasmid replicon, with 18.2% (8/44) of those containing three or more different plasmid replicons (Table 1 and Figure 1). The replicons detected in the *S.* Typhimurium strains analyzed were Col4401, IncX1, IncFIB(S), IncFII(S), and IncFII(pCoo). The plasmids found in *S.* Derby included Col8282, IncB/O/K/Z, IncFIB (H89-PhagePlasmid), IncQ1, IncI1-I (Alpha), and Col(pHAD28), as shown in Table 1. A total of 10 of the 44 isolates (22.7%) had the IncFIB(H89-PhagePlasmid) found in 2 *S.* Hato and 8 *S.* Derby isolates. The isolates containing this replicon all had sequences with 99% identity to AnCo3, a phage-like plasmid detected in a clinical *S*. Derby isolate from Canada. However, in one *S*. Hato isolate, the sequence was distributed over many contigs [12]. All IncB/O/K/Z replicons identified were partial sequences. IncQ1 replicons were partial sequences in 21 isolates.

Plasmid replicons and AMR genes were present on the same contig in 25 isolates (Table 2). However, other AMR genes were likely physically linked to plasmid replicons but were not detected in the WGS assembly as evidenced by some isolates containing AR genes and plasmid replicons with strong positive correlations (r > 0.7) for the co-occurrence of AR genes and plasmid replicons. In particular, the IncQ1 replicon was strongly correlated with *tet*A, *sul*2, *dfr*A14, and multiple aminoglycoside resistance genes (Figure 2 and Appendix A).

### 2.4. Phylogenetic Analysis of S. Derby Isolates 

Both *S*. Derby isolates containing the *fos*A7 gene were compared to publicly available *S.* Derby genomes from chicken sources using cgMLST (Figure 3). The two isolates from this study were the only members of cgMLST type 227,637 in the analyzed dataset and were located on a branch by themselves. 

## 3. Discussion

The present study shows that poultry is a reservoir of MDR *Salmonella* strains in Burkina Faso. Kagambèga et al. [4] previously reported this. A previous study demonstrated that Burkina Faso does not have a commercial slaughterhouse for chickens and that slaughtering is instead carried out at traditional markets [9]. The conditions of chicken slaughter do not respect good hygienic practices, and this undoubtedly promotes cross-contamination, especially during evisceration, between carcasses and chicken feces, representing a risk to human health in Burkina Faso.

MDR *Salmonella* Typhimurium and Kentucky regularly cause human salmonellosis in Burkina Faso, but the sources of these salmonellosis cases remain uninvestigated [13,14]. Many researchers have demonstrated that poultry eggs and meat are major vehicles for human salmonellosis, which is exacerbated by imports from around the world [4,15,16]. The uncontrolled use of antibiotics in Burkina Faso poultry farming contributes to the development of AMR in pathogens and commensal flora. Moreover, the emergence of AMR in bacteria from poultry farms has generated human health concerns due to the consumption of contaminated meat and eggs [17]. 

*Salmonella* Hato and Derby were the most prevalent serotypes isolated from chickens in this study. Previous studies in Burkina Faso revealed that *S.* Derby and *S.* Hato were the major serotypes circulating in poultry [9,10]. Abdelkader et al. [18] found similar results in Niger, which borders Burkina Faso. *S*. Hato and *S.* Derby isolates showed phenotypic resistance to different classes of antibiotics, such as aminoglycosides, tetracycline, trimethoprim, and sulfonamides, and contained genes predicted to confer resistance to fosfomycin. Interestingly, eight *S*. Derby and two *S*. Hato isolates contained a phage-like plasmid with 99% identity to AnCo3. Phage-like plasmids have previously been described in North America to be associated with *bla*_CTX-M-15_ genes. AnCo3 was first identified in a *S*. Derby clinical isolate in Canada. Although similar to AnCo and AnCo2, which contain *bla*_CTX-M-15_, this phage-like plasmid contains no AMR genes. To the best of our knowledge, this is the first report of a phage-like plasmid in Burkina Faso, indicating the global spread of these emerging mobile genetic elements.

However, several of the AMR genes identified in *S.* Hato isolates were only partial sequences. Despite the genes for sulfonamide and trimethoprim resistance appearing as partial in the WGS data, most of these isolates still showed phenotypic resistance to trimethoprim/sulfamethoxazole. These isolates also contained partial IncQ1 replicon sequences, some of which were co-located on contigs with IncI1 replicons. It is possible that these plasmids have merged and that the AMR genes carried have been disrupted. It is also possible that these isolates contained yet unknown resistance genes for these antibiotics. However, it is also possible that an assembly error stemming from repeated DNA sequences, which is notoriously difficult to assemble, caused these genes to appear partial when they are in fact not. In all of these cases of partial sequences, long-read sequencing would be beneficial to further investigate the genetic structure of these plasmids and AMR genes.

Unsurprisingly, the *S.* Derby isolates containing the *fos*A7 gene in this study were genetically unique as compared to publicly available sequences from chickens. The majority of the publicly available *Salmonella* genomes are from countries with surveillance systems, which Burkina Faso lacks, so it is reasonable that these geographically distinct isolates would also be genetically unique.

*Salmonella* Typhimurium isolated in this study possessed *bla*_TEM-1B_ β-lactamase for β-lactam resistance. Extended-spectrum β-lactam resistance could not be confirmed on the susceptibility panel. One *S*. Typhimurium isolate possessed the resistance gene *mph*(A) for macrolide resistance, which could not be confirmed because the panel lacked a macrolide antibiotic. MDR *S*. Typhimurium strains were previously isolated from chickens, and they showed more than 80% genetic similarity to *S.* Typhimurium isolated from human patients [4]. These facts show that chickens and their products constitute a potential danger for the colonization of humans with antibiotic-resistant *Salmonella*. In this study, 36.3% (37/102) of the strains analyzed contained resistance genes for two or more antibiotic classes, with resistance genes from the aminoglycoside class being the most prevalent. Some strains contained up to nine different resistance genes. 

The resistance genes found in this study did not always correlate with the resistance phenotypes. While many isolates possess aminoglycoside resistance genes, the antimicrobial susceptibility test results showed that only four (3.9%) isolates were resistant to gentamicin and that none were resistant to tobramycin. One *S.* Typhimurium isolate only possessed the *aac*(6′)-Iaa gene but showed phenotypic resistance to tetracycline and trimethoprim/sulfamethoxazole. In this case, it is possible that the strain lost a plasmid containing the genes for resistance to these drugs between susceptibility testing and sequencing or that the isolate contains yet unknown genes.

Several antibiotic resistance genes were also detected where phenotypic resistance was not confirmed. For example, there is no CLSI method for phenotypic determination of fosfomycin resistance using broth microdilution, so phenotypic resistance could not be determined. Additionally, AAC(6′) enzymes inactivate aminoglycoside antibiotics by acetylating their substrates at the 6′ position and can confer resistance to amikacin and kanamycin, which were not included on the panel of antibiotics used [19]. However, the *aac*(6′)-Iaa gene has been demonstrated to be non-functional in *Salmonella* unless the strain possesses a mutation to render the promotor for the gene functional [20].

Three of the four *S*. Kentucky isolates identified in poultry in this study showed multi-drug resistance phenotypically, with nine different resistance genes conferring resistance to four or more different antibiotic classes and the *qac*E gene conferring resistance to antiseptics, although this gene was only partially present. In contrast, a study conducted by Chuanchuen et al. [21] in Thailand found that twenty-seven percent of the *Salmonella* strains isolated from poultry and swine possessed *qac*EΔ1 and that none of them harbored *qac*E. This *qac*E identified in this study could be explained by the repeated usage of disinfectants, including quaternary ammonium compounds (QACs), in the farm environment in Burkina Faso. This may increase the selection and persistence of bacteria with reduced susceptibility not only to antiseptics but also possibly to antibiotics [22]. However, phenotypic resistance would have to be confirmed as only a partial gene is present.

The use of PlasmidFinder in this study detected the plasmids with replicon sequences IncFIB(S), IncFII(S), IncFII(pCoo), IncFIB (H89-PhagePlasmid), IncB/O/K/Z, IncX1, and IncQ1 in *Salmonella* isolates. These isolates carried resistances genes for four, three, and two/one classes of antibiotics, including aminoglycosides, β-lactams, sulfonamides, tetracyclines, and phenicols. Villa et al. [23] reported similar results on the IncF group carrying ESBL or plasmid-mediated quinolone or aminoglycoside resistance genes. Carattoli et al. [24] demonstrated that the IncF plasmid family is prevalent in clinically resistant isolates of Enterobacteriaceae. Moreover, IncF can be virulence-associated plasmids, which give host bacteria the ability to cause a more virulent infection [25]. IncX1 plasmids have been associated with genes for resistance to β-lactams and aminoglycosides in *Salmonella* isolated in the USA and genes for resistance to quinolones globally [26,27]. Isolates carrying IncQ1 replicons also carried aminoglycoside resistance genes and *sul*2 with a strong correlation. This result was not surprising, as IncQ1 plasmids are known to be commonly associated with genes for resistance to aminoglycosides, tetracyclines, and sulfonamides [28]. 

The present study concurs with previous research that the T57S substitution detected in *par*C is not always associated with a quinolone resistance phenotype since it has been found in both resistant and susceptible isolates [11,29]. Feng et al. [30] also found similar results in a study of a *Salmonella* Goldcoast lineage in Northern Taiwan, where a single T57S mutation was not always sufficient to confer clinically significant resistance. 

## 4. Materials and Methods

### 4.1. Bacterial Strains

Isolates were collected during a previous investigation of *Salmonella* found in various foods, food animals, and humans in Burkina Faso [9]. The *Salmonella* isolates (*n* = 102) from the cecal and/or intestinal contents of slaughtered chickens used in this study were obtained from the Laboratoire de Biologie Moléculaire, d’épidémiologie et de surveillance des bactéries et virus transmissible par les aliments (LaBESTA)/Université Joseph KI-ZERBO, Burkina Faso. Slaughtered chickens were sourced from markets in different villages across the country. *Salmonella* were isolated using standard methods as previously described [9].

### 4.2. Antimicrobial Susceptibility Testing

For antibiotic susceptibility testing, the isolates were streaked onto Tryptic Soy Agar (TSA) with 5% sheep blood (BBL, Fisher Scientific, Pittsburg, PA, USA) and incubated for 24 h at 37 °C. One colony from each plate was streaked onto a new TSA blood plate for another 24 h at 37 °C. Susceptibility testing was performed using broth microdilution, following the manufacturer’s instructions for the Sensititre™ semi-automated antimicrobial susceptibility system (TREK Diagnostic Systems Inc., Cleveland, OH, USA) and the Sensititre™ Gram-Negative plate format, with plate code GN4F (Thermo, Fisher Scientific, Pittsburg, PA, USA). Minimum inhibitory concentrations (MICs, µg/mL) of all *Salmonella* isolates were classified as resistant, intermediate, or susceptible to the antimicrobials tested using the breakpoints set by the Clinical and Laboratory Standards Institute (CLSI) [31], with the exception of tigecycline. A breakpoint for resistance to tigecycline for Enterobacteriaceae has not been defined, and, therefore, we did not make a judgement on tigecycline resistance. Antimicrobial breakpoints were as follows: Amikacin (≥64 µg mL^−1^); Piperacillin/tazobactam (≥128/4 µg mL^−1^); Ticarcillin/clavulanic acid (≥128/2 µg mL^−1^); Levofloxacin (≥2 µg mL^−1^); Nitrofurantoin (≥128 µg mL^−1^); Tetracycline (≥16 µg mL^−1^); Doripenem (≥4 µg mL^−1^); Minocycline (≥16 µg mL^−1^); Ertapenem (≥2 µg mL−1); trimethoprim/sulfamethoxazole (≥4/76 µg mL^−1^); Imipenem (≥4 µg mL^−1^); Piperacillin (≥128 µg mL^−1^); Meropenem (≥4 µg mL^−1^); gentamicin (≥16 µg mL^−1^); Cefazolin (≥32 µg mL^−1^); Tobramycin (≥16 µg mL^−1^); Ceftazidime (≥16 µg mL^−1^); Ampicillin/sulbactam (≥32/16 µg mL^−1^); Aztreonam (≥16 µg mL^−1^); Ampicillin (≥32 µg mL^−1^); Cefepime (≥16 µg mL^−1^); Ciprofloxacin (≥1 µg mL^−1^); and Ceftriaxone (≥4 µg mL^−1^). For the analysis, isolates identified as intermediate were considered susceptible to the drug. Control strains used were *E. coli* ATCC 25922, *Pseudomonas aeruginosa* ATCC 27853, *Enterococcus faecalis* ATCC 29212, and *Staphylococcus aureus* ATCC 29213. For each isolate, a final inoculum of 1.5 × 10^8^ CFU/mL was targeted. The panels were read after 18 h of incubation at 35 °C.

### 4.3. DNA Extraction, Whole-Genome Sequencing, Assembly, Annotation, and Molecular Serotyping

DNA extraction, library preparation, whole-genome sequencing, assembly, and annotation for the 102 *Salmonella* strains were completed as previously reported [11]. Briefly, libraries were prepared using Nextera XT DNA library preparation kits, which were sequenced using either a 300 or 500 cycle Illumina MiSeq version 2 reagent kit. Reads were assembled using A5 and annotated with the NCBI Prokaryotic Genome Annotation Pipeline [32]. The sequences were deposited into NCBI under BioProject no. PRJNA679582 (https://www.ncbi.nlm.nih.gov/bioproject/PRJNA679582 accessed on 22 March 2022). The serovar determination of the strains using SeqSero was previously described [11]. Identification of serotypes, antibiotic resistance genes, chromosomal mutations, MLST, and plasmids was carried out.

Antibiotic resistance genes and chromosomal point mutations associated with resistance were identified using ResFinder 4.1 through the Center for Genomics Epidemiology (CGE) website (https://cge.cbs.dtu.dk/services/ accessed on 6 May 2022) [33]. Genes with 80% identity and greater than 90% coverage were considered present, and genes with coverage between 40% and 90% were considered present but partial genes. Complete but disrupted genes were noted. Multilocus sequence type (MLST) for each isolate was identified using MLST 2.0 through CGE [34]. PlasmidFinder 2.1 accessed through CGE was used to identify plasmid replicons [35,36]. Replicons with 80% identity and greater than 90% coverage were considered present, and replicons with coverage between 40% and 90% were considered present but partial replicons. Partial AMR genes and replicons were confirmed as partial or complete but disrupted using BLAST [37]. For incompatibility groups with an established scheme, plasmid MLST (pMLST) type was determined using https://pubmlst.org/organisms/plasmid-mlst/ (accessed on 6 May 2022) [38]. Plasmid replicons or pMLST gene targets found on the same contig as AMR genes were noted. Contigs containing phage-like plasmid replicons were confirmed as phage-like plasmids with BLAST.

### 4.4. Phylogenetic Analysis of Salmonella Derby Isolates

Raw paired-end fastq files of both *S.* Derby isolates (S171 and S251), which contained *fos*A7, were imported into Enterobase (https://enterobase.warwick.ac.uk/, accessed on 6 May 2022) and compared to all the publicly available genomes of *S.* Derby (*n* = 197) sourced from poultry in Enterobase, updated on 12 October 2021, using single-nucleotide polymorphisms (SNPs) and hierarchical clustering of core genome (cg) MLST (HierCC) (Zhou et al., 2020). Our study isolates (S171 and S251) and the retrieved genomes from Enterobase were all aligned to the reference *S.* Derby 2014LSAL01779 complete genome (CP026609.1) and designated to HC100 differing by ≤100 core genomic alleles. 

### 4.5. Statistical Analysis

To determine the overall distribution of plasmid replicon types and antimicrobial resistance genes among the examined *Salmonella* serovars, a heatmap with hierarchical clustering was generated using package “pheatmap” in R software (version 3.4.2). A correlation analysis was also performed to determine the association of both determinants among the examined *Salmonella* isolates. Antimicrobial resistance genes and plasmid replicons results, including partial sequences, were converted into binary data (0/1), where the presence of plasmid replicons and resistance genes in isolates received scores of 1, whereas absence of both determinants received scores of 0. The binary data (0/1) for antimicrobial resistance genes and plasmid replicon types were uploaded into R software (version 3.6.1; https://www.r-project.org, accessed on 6 May 2022), and the correlation was calculated at a significance of *p* < 0·05 using “cor” and “cor.mtest” functions. The correlation plot was then generated using the “corrplot” function. Based on the values of r, the degree of correlation is considered strong, moderate, and weak if r value is >0.6, 0.4–0.6, and <0.4, respectively.

## 5. Conclusions

This study shows once again that chicken constitutes a reservoir not only of pathogenic bacteria but also of bacteria that are multi-drug resistant. Chicken is a good source of animal protein and is very popular in Burkina Faso, as production is expanding in the country. This food-producing animal is a reservoir of multi-drug-resistant *Salmonella*, but this study is the first one in the country that reports potential resistance to antiseptics associated with multi-drug resistance. Unfortunately, the country does not have suitable slaughterhouses for chickens, and each market designates one site to slaughter, sell, and roast chickens, which very often smells foul and is visibly unsanitary. The soil and detritus from these poultry processing sites are transported by rainwater to the environment, which contributes to the pollution of water reservoirs and the environment with resistant pathogenic bacteria. Authorities should consider implementing a farm-to-fork quality control system to minimize the risk of pathogenic bacteria contaminating chickens and, subsequently, consumers. Using WGS is a very quick solution to characterize the genome of pathogenic bacteria. Efforts must be made to popularize these methods in developing countries such as Burkina Faso.

## Figures and Tables

**Figure 1 antibiotics-11-00782-f001:**
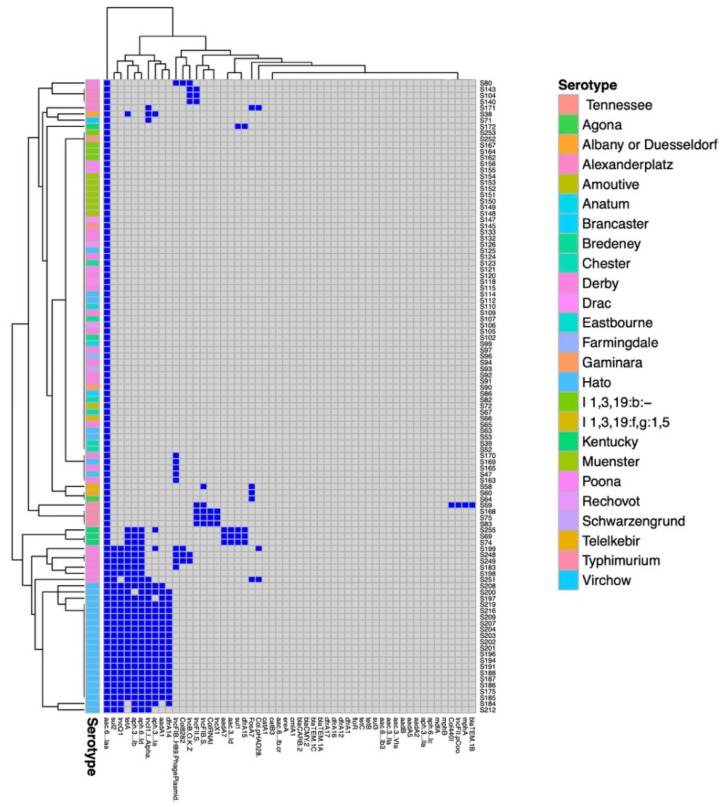
Antibiotic resistance genes and plasmid replicons present in *Salmonella* isolated from slaughtered chickens in Burkina Faso. Genes and replicons present are indicated by dark blue squares; absent genes and replicons are indicated by gray squares. The serotypes are indicated by colored blocks as defined in the key. Relationships based on presence/absence of these genetic elements are indicated by the supporting dendrograms.

**Figure 2 antibiotics-11-00782-f002:**
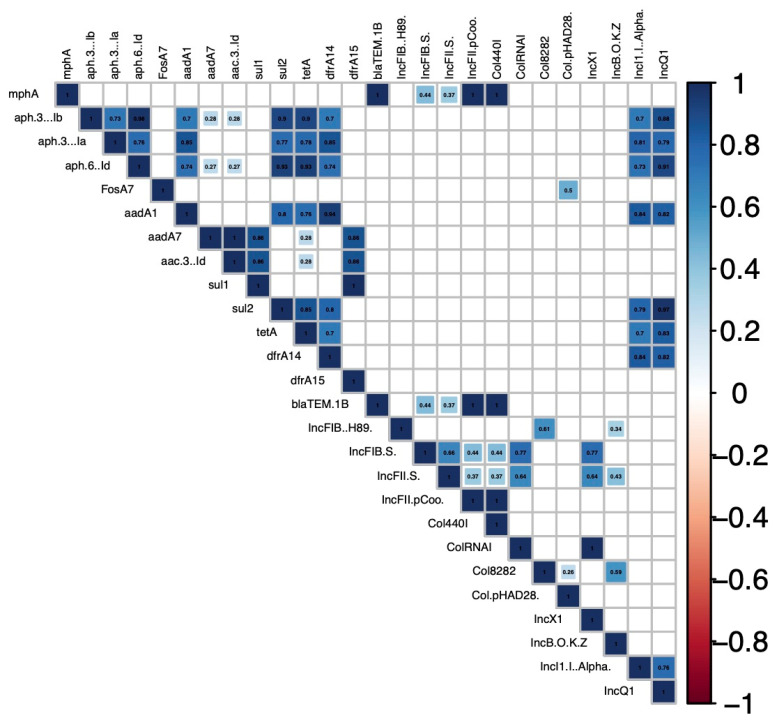
Correlation coefficients for whole and partial antibiotic resistance genes and plasmid replicons present in *Salmonella* isolates from slaughtered chickens. The blue colors of boxes indicate positive correlation with significance calculated at *p* < 0.05. The strength of color corresponds to the numerical value of the correlation coefficient (*r*). Blank boxes indicate non-significant correlations.

**Figure 3 antibiotics-11-00782-f003:**
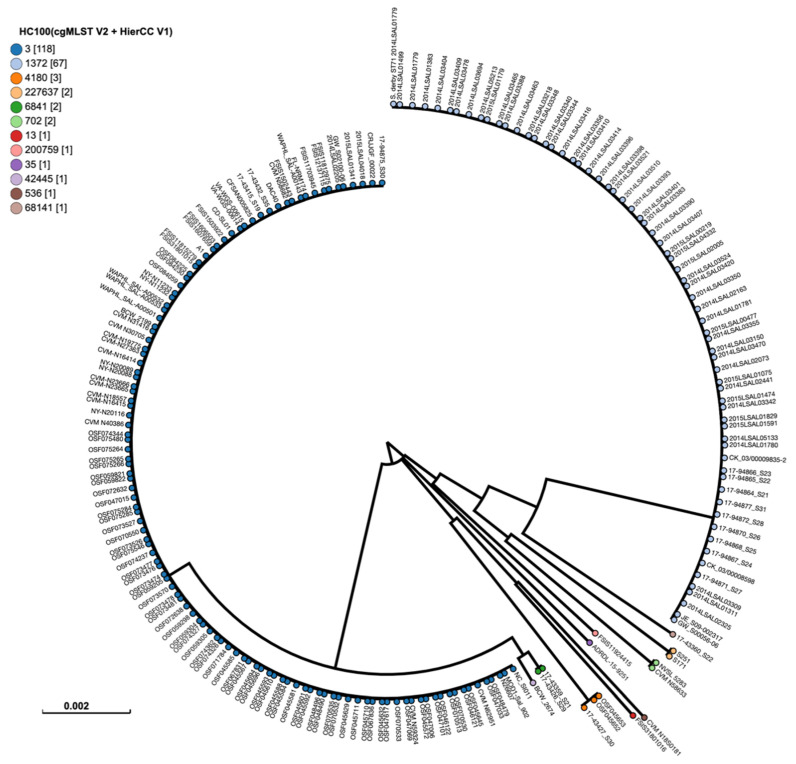
Phylogenetic analysis of the examined *S.* Derby isolates containing *fos*A7 (*n* = 2) and publicly available *S*. Derby isolates from chicken in Enterobase (https://enterobase.warwick.ac.uk/ accessed on 1 June 2022) using single-nucleotide polymorphisms (SNPs) and hierarchical clustering of core genome (cg) MLST (HierCC). The legend shows cgMLST HC100, which indicates allelic differences of no more than 100 of 2850 core genomic alleles among isolates.

**Table 1 antibiotics-11-00782-t001:** Characteristics of *Salmonella* isolated from slaughtered chickens.

Sample	Serotype	Antimicrobial Resistances Genes ^a^	Phenotypic Resistance Profile ^b,c^	Plasmid Replicons ^c^	MLST
S38	Albany or Dusseldorf	*aac*(6′)-Iaa; *aph*(3′)-Ia; *tet*(A)	TET	IncI1-I (Alpha)	292
S39	Chester	*aac*(6′)-Iaa	ND	ND	411
S47	Hato	*aac*(6′)-Iaa	ND	IncFIB (H89-PhagePlasmid)	Unknown
S52	Chester	*aac*(6′)-Iaa	ND	ND	411
S53	Hato	*aac*(6′)-Iaa	ND	ND	3899
S58	Telelkebir	*aac*(6′)-Iaa; *fos*A7	ND	IncFIB(S)	2386
S59	Typhimurium	*aac*(6′)-Iaa; *bla*_TEM-1B_; *mph*(A)	AMP; AMPSUL (A/S2); PIP; TICCLA(TIM2)	Col440I, IncFIB(S), IncFII(S), IncFII(pCoo)	313
S60	Telelkebir	*aac*(6′)-Iaa; *fos*A7	ND	ND	5494
S63	Hato	*aac*(6′)-Iaa	ND	ND	3899
S64	Agona	*aac*(6′)-Iaa; *fos*A7	ND	ND	7876
S65	Derby	*aac*(6′)-Iaa	ND	ND	7119
S66	I 1,3,19:f,g:1,5	*aac*(6′)-Iaa	ND	ND	Unknown
S67	Chester	*aac*(6′)-Iaa	ND	ND	411
S69	Kentucky	*aac*(6′)-laa; *aac*(3)-Id; *aad*A7; *aph*(3″)-Ib; *aph*(6)-Id; *dfr*A15; *sul*1; *tet*(A)	GEN; TET; TRISUL(SXT)	ND	314
S71	Virchow	*aac*(6′)-Iaa	ND	IncI1-I (Alpha)	181
S72	Amoutive	*aac*(6′)-Iaa	ND	ND	Unknown
S74	Kentucky	*aac*(6′)-Iaa; *aac*(3)-Id; *aad*A7; *aph*(3″)-Ib; *aph*(6)-Id; *dfr*A15; *sul*1; *tet*(A)	GEN; TET; TRISUL(SXT)	ND	314
S75	Typhimurium	*aac*(6′)-Iaa	GEN; TET; TRISUL(SXT)	ColRNAI, IncFIB(S), IncFII(S), IncX1	19
S80	Derby	*aac*(6′)-Iaa	ND	Col8282, IncFIB (H89-PhagePlasmid)	5421
S82	Chester	*aac*(6′)-Iaa	ND	ND	411
S83	Typhimurium	*aac*(6′)-Iaa	ND	ColRNAI, IncFIB(S), IncFII(S), IncX1	19
S86	Brancaster	*aac*(6′)-Iaa	ND	ND	Unknown
S90	Gaminara	*aac*(6′)-Iaa	ND	ND	2152
S91	Derby	*aac*(6′)-Iaa	ND	ND	7882
S92	Derby	*aac*(6′)-Iaa	ND	ND	7882
S93	Schwarzengrund	*aac*(6′)-Iaa	ND	ND	96
S94	Derby	*aac*(6′)-Iaa	ND	ND	7880
S96	Farmingdale	*aac*(6′)-Iaa	ND	ND	Uknown
S97	Derby	*aac*(6′)-Iaa	ND	ND	7880
S99	Anatum	*aac*(6′)-Iaa	ND	ND	5197
S102	Bredeney	*aac*(6′)-Iaa	ND	ND	306
S104	Alexanderplatz	*aac*(6′)-Iaa	ND	IncFII(S)	Unknown
S105	Derby	*aac*(6′)-Iaa	ND	ND	7882
S106	Rechovot	*aac*(6′)-Iaa	ND	ND	Unknown
S107	Bredeney	*aac*(6′)-Iaa	ND	ND	306
S109	Derby	*aac*(6′)-Iaa	ND	ND	7882
S110	Eastbourne	*aac*(6′)-Iaa	ND	ND	414
S112	Hato	*aac*(6′)-Iaa	ND	ND	3997
S114	Hato	*aac*(6′)-Iaa	ND	ND	Unknown
S115	Derby	*aac*(6′)-Iaa	ND	ND	7880
S118	Poona	*aac*(6′)-Iaa	ND	ND	308
S120	Derby	*aac*(6′)-Iaa	ND	ND	7882
S121	Poona	*aac*(6′)-Iaa	ND	ND	308
S123	Bredeney	*aac*(6′)-Iaa	ND	ND	306
S124	Derby	*aac*(6′)-Iaa	ND	ND	7880
S125	Hato	*aac*(6′)-Iaa	ND	ND	Unknown
S126	Rechovot	*aac*(6′)-Iaa	ND	ND	Unknown
S132	Derby	*aac*(6′)-Iaa	ND	ND	7882
S133	Derby	*aac*(6′)-Iaa	ND	ND	7882
S140	Alexanderplatz	*aac*(6′)-Iaa	ND	IncFII(S)	Unknown
S143	Alexanderplatz	*aac*(6′)-Iaa	ND	IncFII(S)	Unknown
S145	Tennessee	*aac*(6′)-Iaa	ND	ND	8398
S147	Drac	*aac*(6′)-Iaa	ND	ND	2221
S148	Muenster	*aac*(6′)-Iaa	ND	ND	321
S149	Muenster	*aac*(6′)-Iaa	ND	ND	321
S150	Muenster	*aac*(6′)-Iaa	ND	ND	321
S151	Muenster	*aac*(6′)-Iaa	ND	ND	321
S152	Muenster	*aac*(6′)-Iaa	ND	ND	321
S153	Muenster	*aac*(6′)-Iaa	ND	ND	321
S154	Muenster	*aac*(6′)-Iaa	ND	ND	321
S155	Poona	*aac*(6′)-Iaa	ND	ND	608
S156	Poona	*aac*(6′)-Iaa	ND	ND	608
S162	I 1,3,19:b:-	*aac*(6′)-Iaa	ND	ND	Unknown
S163	Derby	*aac*(6′)-Iaa	ND	IncFIB (H89-PhagePlasmid)	3135
S164	I 1,3,19:b:-	*aac*(6′)-Iaa	ND	ND	Unknown
S165	Derby	*aac*(6′)-Iaa	ND	IncFIB (H89-PhagePlasmid)	3135
S167	I 1,3,19:b:-	*aac*(6′)-Iaa	ND	ND	Unknown
S168	Typhimurium	*aac*(6′)-Iaa	ND	ColRNAI, IncFIB(S), IncFII(S), IncX1	19
S169	Hato	*aac*(6′)-Iaa	ND	IncFIB (H89-PhagePlasmid)	3292
S170	Derby	*aac*(6′)-Iaa	ND	IncFIB (H89-PhagePlasmid)	3135
S171	Derby	*aac*(6′)-Iaa; *fos*A7	ND	Col(pHAD28), IncI1-I (Alpha)	7881
S172	Kentucky	*aac*(6′)-Iaa; *dfr*A15; *sul*1	TRISUL(SXT)	ND	314
S175	Hato	*aac*(6′)-Iaa; *aad*A1; *sul*2; *tet*(A)	TET; TRISUL(SXT)	IncI1-I (Alpha)	3899
S183	Derby	*aac*(6′)-Iaa; *aph*(3″)-Ib; *aph*(6)-Id; *sul*2; *tet*(A)	MIN; TET	IncFIB (H89-PhagePlasmid), IncQ1	3135
S184	Hato	*aac*(6′)-Iaa; *aph*(3′)-Ia; [*aph*(3′’)-Ib]; *aph*(6)-Id; *dfr*A14; *sul*2	TRISUL(SXT)	IncI1-I (Alpha)	3899
S185	Hato	*aac*(6′)-Iaa; *aad*A1; *aph*(3′)-Ia; *dfr*A14; *sul*2; *tet*(A)	MIN; TET; TRISUL(SXT)	IncI1-I (Alpha)	3899
S186	Hato	*aac*(6′)-Iaa; *aad*A1; *sul*2; *tet*(A)	TET; TRISUL(SXT)	IncI1-I (Alpha)	3899
S187	Hato	*aac*(6′)-Iaa; *aad*A1; *aph*(3′)-Ia; *aph*(6)-Id; *sul*2; *tet*(A)	TET; TRISUL(SXT)	IncI1-I (Alpha)	3899
S188	Hato	*aac*(6′)-Iaa; *aad*A1; *aph*(3′)-Ia; [*aph*(3″)-Ib]; *sul*2; *tet*(A)	MIN; TET; TRISUL(SXT)	IncI1-I (Alpha)	3899
S191	Hato	*aac*(6′)-Iaa; *aad*A1; *aph*(3′)-Ia; *aph*(6)-Id; *dfr*A14; *sul*2; *tet*(A)	MIN; TET; TRISUL(SXT)	IncI1-I (Alpha)	3899
S194	Hato	*aac*(6′)-Iaa; *aad*A1; *aph*(3′)-Ia; [*aph*(3″)-Ib]; *sul*2; *tet*(A)	TET; TRISUL(SXT)	IncI1-I (Alpha)	3899
S196	Hato	*aac*(6′)-Iaa; *aad*A1; *aph*(3′)-Ia; [*aph*(3″)-Ib]; *sul*2; *tet*(A)	TET; TRISUL(SXT)	IncI1-I (Alpha)	3899
S197	Hato	*aac*(6′)-Iaa; *aad*A1; *sul*2; *tet*(A)	TET; TRISUL(SXT)	IncI1-I (Alpha)	3899
S198	Derby	*aac*(6′)-Iaa; *aph*(3′’)-Ib; *aph*(6)-Id; *sul*2; *tet*(A)	MIN; TET	IncQ1	3135
S199	Derby	*aac*(6′)-Iaa; *aph*(3′’)-Ib; *aph*(6)-Id; *sul*2; *tet*(A)	MIN; TET	Col(pHAD28), Col8282, IncFIB (H89-PhagePlasmid), IncQ1	3135
S200	Hato	*aac*(6′)-Iaa; *aad*A1; *aph*(3′)-Ia; *aph*(6)-Id; *dfr*A14; *sul*2; *tet*(A)	TET; TRISUL(SXT)	IncI1-I (Alpha)	3899
S201	Hato	*aac*(6′)-Iaa; *aad*A1; *aph*(3′)-Ia; [*aph*(3″)-Ib]; *aph*(6)-Id; *dfr*A14; *sul*2; *tet*(A)	TRISUL(SXT)	IncI1-I (Alpha)	3899
S202	Hato	*aac*(6′)-Iaa; *aad*A1; *aph*(3′)-Ia; [*aph*(3″)-Ib]; *aph*(6)-Id; *dfr*A14; *sul*2; *tet*(A)	TET; TRISUL(SXT)	IncI1-I (Alpha)	3899
S203	Hato	*aac*(6′)-Iaa; *aad*A1; *aph*(3′)-Ia; *sul*2; *tet*(A)	TET; TRISUL(SXT)	IncI1-I (Alpha)	3899
S204	Hato	*aac*(6′)-Iaa; *aad*A1; *aph*(3′)-Ia; *aph*(6)-Id; *sul*2; *tet*(A)	TET; TRISUL(SXT)	IncI1-I (Alpha)	3899
S207	Hato	*aac*(6′)-Iaa; *aad*A1; *aph*(3′)-Ia; [*aph*(3″)-Ib]; *aph*(6)-Id; *dfr*A14; *sul*2; *tet*(A)	TET; TRISUL(SXT)	IncI1-I (Alpha)	3899
S208	Hato	*aac*(6′)-Iaa; *aad*A1; *aph*(3′)-Ia; *sul*2; *tet*(A)	TET; TRISUL(SXT)	IncI1-I (Alpha)	3899
S209	Hato	*aac*(6′)-Iaa; *aad*A1; *aph*(3′)-Ia; *sul*2; *tet*(A)	TET; TRISUL(SXT)	IncI1-I (Alpha)	3899
S212	Hato	*aac*(6′)-Iaa; *aph*(3′’)-Ib; *aph*(6)-Id; *sul*2	ND	IncI1-I (Alpha)	3899
S216	Hato	*aac*(6′)-Iaa; *aad*A1; *aph*(3′)-Ia; *sul*2; *tet*(A)	TET; TRISUL(SXT)	IncI1-I (Alpha)	3899
S219	Hato	*aac*(6′)-Iaa; *aad*A1; *aph*(3′)-Ia; [*aph*(3″)-Ib]; *sul*2; *tet*(A)	TET; TRISUL(SXT)	IncI1-I (Alpha)	3899
S248	Derby	*aac*(6′)-Iaa; *aph*(3′’)-Ib; *aph*(6)-Id; *sul*2; *tet*(A)	TET	Col8282, IncFIB (H89-PhagePlasmid), IncQ1	Unknown
S249	Derby	*aac*(6′)-Iaa; *aph*(3′’)-Ib; *aph*(6)-Id; *sul*2; *tet*(A)	TET	Col8282, IncFIB (H89-PhagePlasmid), IncQ1	Unknown
S251	Derby	*aac*(6′)-Iaa; *aph*(3″)-Ib; *aph*(6)-Id; *fos*A7; *sul*2; *tet*(A)	MIN; TET	Col(pHAD28), IncI1-I (Alpha)	7881
S252	Tennessee	*aac*(6′)-Iaa	ND	ND	8398
S253	I 1,3,19:b:-	*aac*(6′)-Iaa	ND	ND	Unknown
S255	Kentucky	*aac*(6′)-Iaa; *aac*(3)-Id; *aad*A7; *aph*(3″)-Ib; *aph*(6)-Id; *dfr*A15; *sul*1; *tet*(A)	GEN; TET; TRISUL(SXT)	ND	314

^a^ Genes in brackets [] are complete but disrupted by an insertion. ^b^ Abbreviations: TICCLA (TIM2), Ticarcillin/clavulanic acid; TET, tetracycline; MIN, Minocycline; TRISUL(SXT), trimethoprim/sulfamethoxazole; PIP, Piperacillin; GEN, gentamicin; AMPSUL (A/S2), Ampicillin/sulbactam; AMP, Ampicillin. Resistance was determined by susceptibility testing using MIC cut-offs from CLSI for resistance to the antibiotics indicated. ^c^ ND indicates “not detected”.

**Table 2 antibiotics-11-00782-t002:** Plasmid replicons physically linked to antibiotic resistance genes.

Isolate	Serotypes	Plasmid Replicon	pMLST Type ^a^	Antibiotic Resistance Genes
S38	Albany or Dusseldorf	IncI1	IncI1 ST 12, CC-12	*aph*(3′)-Ia; *tet*(A)
S175	Hato	IncI1	IncI1 ST 12, CC-12	*aph*(3″)-Ib; *aph*(6)-Id; *dfr*A14; *sul*2
S183	Derby	IncQ1	NA	*aph*(3″)-Ib; *aph*(6)-Id; *sul*2; *tet*(A)
S184	Hato	IncI1 and IncQ1	IncI1 ST 12, CC-12	*aph*(3′’)-Ib; *aph*(6)-Id; *dfr*A14; *sul*2
S185	Hato	IncI1	IncI1 ST 12, CC-12	*aph*(3″)-Ib; *aph*(6)-Id
S186	Hato	IncI1	IncI1 ST 12, CC-12	*aph*(3″)-Ib; *aph*(6)-Id; *dfr*A14; *sul*2
S187	Hato	IncQ1	NA	*aph*(3″)-Ib; *aph*(6)-Id; *dfr*A14; *sul*2;
S188	Hato	IncI1	IncI1 ST 12, CC-12	*aph*(3″)-Ib; *aph*(6)-Id
S194	Hato	IncI1	IncI1 ST 12, CC-12	*aph*(6)-Id
S196	Hato	IncI1	IncI1 ST 12, CC-12	*aph*(3″)-Ib; *aph*(6)-Id; *dfr*A14
S197	Hato	IncI1	IncI1 ST 12, CC-12	*aph*(3″)-Ib; *aph*(6)-Id; *dfr*A14; *sul*2
S198	Derby	IncQ1	NA	*aph*(3′’)-Ib; *aph*(6)-Id; *sul*2; *tet*(A)
S199	Derby	IncQ1	NA	*aph*(3′’)-Ib; *aph*(6)-Id; *sul*2; *tet*(A)
S201	Hato	IncI1 and IncQ1	IncI1 ST 12, CC-12	*aph*(3″)-Ib; *aph*(6)-Id; *dfr*A14; *sul*2
S202	Hato	IncI1 and IncQ1	IncI1 ST 12, CC-12	*aph*(3″)-Ib; *aph*(6)-Id; *dfr*A14; *sul*2
S203	Hato	IncI1	IncI1 ST 12, CC-12	*aph*(3″)-Ib; *aph*(6)-Id; *dfr*A14; *sul*2
S204	Hato	IncQ1	NA	*aph*(3″)-Ib; *aph*(6)-Id; *dfr*A14; *sul*2
S207	Hato	IncI1 and IncQ1	IncI1 ST 12, CC-12	*aph*(3″)-Ib; *aph*(6)-Id; *dfr*A14; *sul*2
S208	Hato	IncI1	IncI1 ST 12, CC-12	*aph*(3″)-Ib; *aph*(6)-Id; *dfr*A14; *sul*2
S209	Hato	IncI1	IncI1 ST 12, CC-12	*aph*(3″)-Ib; *aph*(6)-Id; *dfr*A14; *sul*2
S212	Hato	IncI1 and IncQ1	IncI1 ST 12, CC-12	*aph*(3′’)-Ib; *aph*(6)-Id; *sul*2
S216	Hato	IncI1	IncI1 ST 12, CC-12	*aph*(3″)-Ib; *aph*(6)-Id; *dfr*A14; *sul*2
S248	Derby	IncQ1	NA	*aph*(3′’)-Ib; *aph*(6)-Id; *sul*2; *tet*(A)
S249	Derby	IncQ1	NA	*aph*(3′’)-Ib; *aph*(6)-Id; *sul*2; *tet*(A)
S251	Derby	Col(pHAD28)	NA	*aph*(3″)-Ib; *aph*(6)-Id; *sul*2; *tet*(A)

^a^ NA indicates “not applicable”, as this replicon type does not have a pMLST scheme.

## Data Availability

The datasets and Accession numbers generated during the current study are available in the NCBI repository under BioProject no. PRJNA679582 (https://www.ncbi.nlm.nih.gov/bioproject/PRJNA679582;https://www.ncbi.nlm.nih.gov/biosample?Db=biosample&DbFrom=bioproject&Cmd=Link&LinkName=bioproject_biosample&LinkReadableName=BioSample&ordinalpos=1&IdsFromResult=679582 accessed on 22 March 2022).

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
