# Peer review of "Resistance Genes, Plasmids, Multilocus Sequence Typing (MLST), and Phenotypic Resistance of Non-Typhoidal Salmonella (NTS) Isolated from Slaughtered Chickens in Burkina Faso"

_antibiotics, 2022, doi:10.3390/antibiotics11060782_

Round 1

Reviewer 1 Report

In this manuscript, the authors presented the results of their study on salmonella isolates in chickens in Burkina Faso.

Although the project is very interesting, I believe it needs to be improved before publication.
In particular, I believe that the results should be better described, especially the Serotypes. A very long figure and table described in 6 lines is not very useful for a reader.

A useful choice in this sense could be to merge results and discussion in one paragraph.

Author Response

Reviewer 1:

In this manuscript, the authors presented the results of their study on salmonella isolates in chickens in Burkina Faso.

Although the project is very interesting, I believe it needs to be improved before publication.
In particular, I believe that the results should be better described, especially the Serotypes. A very long figure and table described in 6 lines is not very useful for a reader.

We thank the reviewer for their time and effort working on our manuscript. We appreciate this remark, as it has identified our poor communication to the reader. Table 1 presents the data collected on the isolates in this study, and not serotype alone. This includes serotype, antimicrobial resistance genes, phenotypic resistance, plasmid replicons, and MLST data. We have changed the headings for the sections in the Results to make this more clear (lines 75, 113, 162, and 184) which now read: Serotypes of isolates from slaughtered chickens, Antimicrobial resistance genes and antibiotic resistance phenotypes detected, Replicon types detected, and Phylogenetic analysis of S. Derby isolates, respectively. We also believe that as it contains most of the data on the isolates, Table 1 is necessary for the paper. However, we have shortened it by removing less important data on parC mutations and partial gene sequences. Table 1 has now been reduced to four pages and we believe this is reasonable to present the data to the reader.

A useful choice in this sense could be to merge results and discussion in one paragraph

We appreciate the reviewer’s suggestion, however we found that the paper was more difficult to understand when these sections were combined, therefore we have left the results and discussion separate sections.

Reviewer 2 Report

The study represents valuable data on antimicrobial resistance of Salmonella's. However presentation is not good. In my opinion, the authors should present results more clearly, the Table 1 is way too big and unnecessary. It should be condensed to present what is really significant. All the results can be given in supplement. Also, the results on MIC determination are almost non presented. This should be shown more clearly. Finally, the conclusions need to be re-written (more detail as follows).

Line 34 –The conclusions given in Chapter 5. are very good, but here in abstract, not. for example: „The results show the risk of eating chicken for consumers in Burkina Faso“ – this is in no way related to detection of antibiotic resistance genes, the risk is the same for every country and is dependent on sanitary conditions. So, this conclusion is not conected to specifically your results. In chapter 5 is better: „The soil and 389 detritus from these poultry processing sites is transported by rainwater to the environ-390 ment, which contributes to the pollution of water reservoirs and the environment with 391 resistant pathogenic bacteria.“

Also, regarding to chapter 5: „Authorities should consider implementing a farm-to-fork quality control system to minimize the risk of multi-drug resistant pathogenic bacteria contamination to consumers.“ – this claim requiring better sanitary control is not just related to AMR genes, but to prevent salmonelosis in general. The same goes to sentence in abstract „Due to the prevalence of these drug resistant patho-35 gens, poultry processing and cooking should be performed with attention to prescribed safe han-36 dling methods to avoid cross-contamination with chicken products.“ You need to clearly separate what did you conclude from your research (on AMR) and what are general guidance for better sanitary conditions.

Line 52 – please state the population of Ouagadougou

Line 66 – „Whole genome sequencing (WGS) allows for the analysis of the Salmonella genome but requires expensive equipment and reagents, which are inaccessible to laboratories in Burkina Faso.“ – please explain the relevance of this statement, I presume this is related to laboratories in the USA, is it some project, collaboration?

Line 87 – Table 1 is way too big. It can be in supplement. In Table 1 you can list maybe several serotypes, or something what you wish to accent, not the whole experimental list.

Line 107 – What do you mean by „phenotypic resistance“?

Line 112 – 139 – This is in my opinion unnecessary listing of data. You should rather focus on the results you find to be important and worthy of explaining in detail. The asme comment is for Lines 155 – 165

Line 176  - I think the Figure 3 is not necessary

Line 303 – please give more detail, not just reference number 9.

Line 304 – There are no results in the manuscript for Antibiotic susceptibility testing, just for WGS. These results should be shown for sure. I see they are listed in table 1 as Phenotypic resistance profile, but this is not good. I believe the resistance profiles should be separated, anyway Table 1 is too big.

Author Response

Reviewer 2:

The study represents valuable data on antimicrobial resistance of Salmonella's. However, presentation is not good. In my opinion, the authors should present results more clearly, the Table 1 is way too big and unnecessary. It should be condensed to present what is significant. All the results can be given in supplement. Also, the results on MIC determination are almost non presented. This should be shown more clearly.

We thank the reviewer for their helpful comments. We agree that Table 1 is too large, and we have reduced it to only four pages, by, as was suggested, by leaving out less important data on partial gene sequences and parC mutations. We believe the data that is left is important enough to leave in the manuscript for the reader to refer too.

We apologize for the confusion about the MICs. As stated in the materials and methods, minimum inhibitory concentrations (MIC, μg/mL) of all Salmonella isolates were classified as resistant, intermediate, or susceptible to the antimicrobials tested using the breakpoints set by the Clinical and Laboratory Standards Institute (These breakpoints are presented in material and method section). Thus, in Table 1, we simply list the abbreviation for the antimicrobials that the isolate indicated was resistant to as defined by CLSI. To make this more clear we have added this information to the foot note on Table 1. Lines 97-98 now reads: “. Resistance was determined by susceptibility testing using MIC cut-offs from CLSI for resistance to the antibiotics indicated.”

 Finally, the conclusions need to be re-written (more detail as follows).

Ok

Line 34 –The conclusions given in Chapter 5. are very good, but here in abstract, not. for example: „The results show the risk of eating chicken for consumers in Burkina Faso“ – this is in no way related to detection of antibiotic resistance genes, the risk is the same for every country and is dependent on sanitary conditions. So, this conclusion is not conected to specifically your results.

Thank you for pointing that out to us, we have corrected it

 In chapter 5 is better: „The soil and 389 detritus from these poultry processing sites is transported by rainwater to the environ-390 ment, which contributes to the pollution of water reservoirs and the environment with 391 resistant pathogenic bacteria.“

Yes, we agree with the reviewer. We want to show the risk link to poultry processing and distribution of pathogenic bacteria in environment (We want to show that the waste from poultry processing can contribute to environment pollution by pathogenic bacteria, because the strains were isolated in chickens during slaughter in market from Burkina Faso, and in this country hygienic conditions are not good, poultry carcass vendors don’t have good management system for waste produced during poultry processing)  

Also, regarding to chapter 5: „Authorities should consider implementing a farm-to-fork quality control system to minimize the risk of multi-drug resistant pathogenic bacteria contamination to consumers.“ – this claim requiring better sanitary control is not just related to AMR genes, but to prevent salmonelosis in general. The same goes to sentence in abstract „Due to the prevalence of these drug resistant patho-35 gens, poultry processing and cooking should be performed with attention to prescribed safe han-36 dling methods to avoid cross-contamination with chicken products.“ You need to clearly separate what did you conclude from your research (on AMR) and what are general guidance for better sanitary conditions.

We agree with the reviewer and have made changes to that section. Lines 404-406 now reads: “Authorities should consider implementing a farm-to-fork quality control system to minimize the risk of pathogenic bacteria contaminating chickens to consumers.”

We agree with the reviewer and have made changes to line 34-37, which now reads: “The results show that chicken can be considered as a reservoir of antibiotic resistant salmonella strains. Due to the prevalence of these drug resistant pathogens, and the potential for food born illnesses, poultry processing and cooking should be performed with attention to prescribed safe handling methods to avoid cross-contamination with chicken products.”

Line 52 – please state the population of Ouagadougou

The population of Ouagadougou was 3,056,000 in 2022. We have added this in line 52.

Line 66 – „Whole genome sequencing (WGS) allows for the analysis of the Salmonella genome but requires expensive equipment and reagents, which are inaccessible to laboratories in Burkina Faso.“ – please explain the relevance of this statement, I presume this is related to laboratories in the USA, is it some project, collaboration?

We apologize for not making this more clear. Burkina -Faso does not have the ability to do WGS analysis, therefore a collaboration with scientists at USDA, ARS in the United States was developed to collect this important data. Strains were isolated in Burkina Faso but WGS cannot be done in Burkina Faso and we uses a collaboration to get it done in USA. Lines 67-71 now read: ” Whole genome sequencing (WGS) allows for the analysis of the Salmonella genome but requires expensive equipment and reagents, which are inaccessible to laboratories in Burkina Faso, therefore, collaborations were developed with scientists at the United States Department of Agriculture, Agricultural Research Service (USDA-ARS).”

Line 87 – Table 1 is way too big. It can be in supplement. In Table 1 you can list maybe several serotypes, or something what you wish to accent, not the whole experimental list.

We have reduced Table 1 to four pages by focusing on the important data presented and discussed in the paper. We believe that because the table presents the data it should remain in the paper.

Line 107 – What do you mean by „phenotypic resistance“?

Sorry for any confusion. Antibiotic Susceptibility testing was performed using broth microdilution following manufacturer’s instructions for the Sensititre™ semi-automated antimicrobial susceptibility system. Which is considered a phenotypic method. Resistance is then scored by comparison of MICs observed to cut-offs defined by the CLSI.

Line 112 – 139 – This is in my opinion unnecessary listing of data. You should rather focus on the results you find to be important and worthy of explaining in detail. The asme comment is for Lines 155 – 165

The authors respectfully disagree with the reviewer that this is unnecessary data listing. We are presenting the data in this fashion to help the reader focus on the important data found in Table 1. This also allows us to compare the data presented from Burkina Faso to data from other locations in the world. This adds the context to data from a country with very little information available.

Line 176  - I think the Figure 3 is not necessary

The authors disagree with the reviewer. The power of WGS is the ability to do this type of comparison which clearly shows the Burkina Faso isolates with fosA are a different phylogenetic group from the rest of the poultry associated S. Derby isolates from other parts of the world. This figure clearly shows this and helps the reader understand just how unique these strains are as compared to the rest of the world.

Line 303 – please give more detail, not just reference number 9

The isolates are from a previous study described in reference 9. Standard laboratory isolation methods were used and are found in reference 9. To clarify this the text was changed to lines 308-315: “Isolates were collected during a previous investigation of Salmonella found in various foods, food animals, and humans in Burkina Faso (9). The Salmonella isolates (n=102) from the cecal and/or intestinal contents of slaughtered chickens used in this study were obtained from the Laboratoire de Biologie Moléculaire, d’épidémiologie et de surveillance des bactéries et virus transmissible par les aliments (LaBES-TA)/Université Joseph KI-ZERBO, Burkina Faso. Slaughtered chickens were sourced from markets in different villages across the country. Salmonella was isolated using standard methods as described previously (9).

Line 304 – There are no results in the manuscript for Antibiotic susceptibility testing, just for WGS. These results should be shown for sure. I see they are listed in table 1 as Phenotypic resistance profile, but this is not good. I believe the resistance profiles should be separated, anyway Table 1 is too big.

We apologize for this misunderstanding. We have added that the MICs were used to score the susceptibility testing based on the CLSI break points and that we only report those that are resistant. Those are then listed in Table 1 where the abbreviation for the antibiotic indicates resistance to that antibiotic in the indicated isolate. This has been added to the footnote of table 1 to make this clear. We believe that Table 1 is the best way to present the susceptibility data as it is directly beside the column listing the resistance genes detected, this way the reader can directly compare phenotypic and genotypic data for the isolates. Line 97-98 now read: “Resistance was determined by susceptibility testing using MIC cut-offs from CLSI for resistance to the antibiotics indicated.”

Reviewer 3 Report

Dear authors

Minor editing and adding new information are suggested to be included in order to improve the reviewed manuscript, as follows:

L3 Please add “paratyphoid” before “Salmonella” and “serotypes” after “Salmonella

L27 Please include “slaughtered” before chickens

L 87 Regarding the row of “S38”, please change from “Duesseldorf” to “Dusseldorf”

L119, L123, and L129 Please add “S.” before “Hato”, “Derby” and “Typhimurium”, respectively.

L137 and L138 Please relocate “serotypes” after “Typhimurium” and “Rechovot”, respectively.

L163 Please add “S.” before “Hato”

L174 Please add a new column including the serotype name of each of the Salmonella isolate.

L177 and L237 Please change from “Salmonella” to “S.”

L251 Please change from “Gentamicin” to “gentamicin”

L258 Please change from “Fosfomycin” to “fosfomycin”

L298-L302 It could be useful if the authors consider including the new information by answering the following suggestions:

  • Period of time during which the study was performed
  • The total number of samples from which the 102 paratyphoid Salmonella isolates were obtained
  • Presence or absence of gross findings related to Salmonella infection among the evaluated carcasses and/or organs
  • Please clarify if “intestinal contents” refers to samples obtained from the small or large intestine

L314 Please add “(CLSI)” after “… Institute”

L381 The authors must emphasize that most of the paratyphoid Salmonella serotypes obtained in broiler chickens in this study are unusual compared to those usually previously obtained from commercial chickens and widely known in the current literature (S. Enteritidis, S. Heidelberg, S. Typhimurium, and, lately, S. Infantis ).  Another point to be added in the Discussion section is the potential link between the paratyphoid Salmonella isolates obtained from slaughtered chickens and those chickens kept during their last weeks of growing at the farm.

An additional suggestion is to include information on the comparison between the MLST obtained in this study and the MLST database of paratyphoid Salmonella from chickens and other affected animal species. This could be valuable information to clarify the potential source/s of these microorganisms for chickens, to link with the same/other geographical locations similarly affected by these Salmonella serotypes, and to propose a bigger picture of the situation intending a “One Health” approach.

Author Response

Reviewer 3:

Dear authors

Minor editing and adding new information are suggested to be included in order to improve the reviewed manuscript, as follows:

The authors thank the review for their helpful suggestions.

L3 Please add “paratyphoid” before “Salmonella” and “serotypes” after “Salmonella

We apologize for this oversight and are glad the reviewer has brought it to our attention. None of the isolates in this study are Paratyphoid or Typhoid Salmonella as they were isolated from chickens. Therefore, we have changed the title and text to indicate that this study is only on Non Typhoidal Salmonella (NTS).

L27 Please include “slaughtered” before chickens

Done, thank you.

L 87 Regarding the row of “S38”, please change from “Duesseldorf” to “Dusseldorf”

Done, thank you for catching that misspelling.

L119, L123, and L129 Please add “S.” before “Hato”, “Derby” and “Typhimurium”, respectively.

Ok, done

L137 and L138 Please relocate “serotypes” after “Typhimurium” and “Rechovot”, respectively.

The authors respectfully disagree with this change as these lists of serotypes are best introduced as serotypes, rather than defining them as serotypes at the end of the list. If the reviewer or the editor has a reason for this change that we are unaware of, we will be happy to make it.

L163 Please add “S.” before “Hato”

Ok, done

L174 Please add a new column including the serotype name of each of the Salmonella isolate.

Ok, done

L177 and L237 Please change from “Salmonella” to “S.”

Ok, done

L251 Please change from “Gentamicin” to “gentamicin”

Ok, done

L258 Please change from “Fosfomycin” to “fosfomycin”,

ok, done

L298-L302 It could be useful if the authors consider including the new information by answering the following suggestions:

Our isolates are from a previously published study of ours (reference 9), therefore the data you request has already been published and we are now publishing the WGS and phenotypic data for the isolates.

  • Period of time during which the study was performed
  • The total number of samples from which the 102 paratyphoid Salmonella isolates were obtained
  • Presence or absence of gross findings related to Salmonella infection among the evaluated carcasses and/or organs these were healthy animals at the time of slaughter.
  • Please clarify if “intestinal contents” refers to samples obtained from the small or large intestine intestinal contents of the large intestine

L314 Please add “(CLSI)” after “… Institute”

ok, done

L381 The authors must emphasize that most of the paratyphoid Salmonella serotypes obtained in broiler chickens in this study are unusual compared to those usually previously obtained from commercial chickens and widely known in the current literature (S. Enteritidis, S. Heidelberg, S. Typhimurium, and, lately, S. Infantis ).  Another point to be added in the Discussion section is the potential link between the paratyphoid Salmonella isolates obtained from slaughtered chickens and those chickens kept during their last weeks of growing at the farm.

That authors appreciate this suggestion, however none of our isolates are paratyphoid, they are all NTS Salmonella enterica serotypes. The point that they are not the usual suspects (Enteritidis, Typhimurium, Heidelberg, Infantis, etc.) is likely due to their origin in Burkina Faso and the serotypes you mention are more like what we see in the USA and the EU. Compared to the other countries in the sub-Saharan area of Africa these serotypes are not that surprising. This is why we are doing the study, there is almost no WGS data from this part of the world and Salmonella is rarely isolated or serotyped when it is, most infections are treated empirically. As these isolates were collected from the large intestinal contents of freshly slaughtered chickens, I would expect that what was isolated would be the same as what the chickens carried on-farm a few weeks prior to slaughter. It would be interesting to do that study though.

An additional suggestion is to include information on the comparison between the MLST obtained in this study and the MLST database of paratyphoid Salmonella from chickens and other affected animal species. This could be valuable information to clarify the potential source/s of these microorganisms for chickens, to link with the same/other geographical locations similarly affected by these Salmonella serotypes, and to propose a bigger picture of the situation intending a “One Health” approach.

We agree that this would help with a one health approach, but there are several reasons this is impractical for the current study. MLST is unfortunately a highly limited typing tool for tracing Salmonella or for trying to do source tracking. More importantly there is no large database of MLST data from this region of the world to compare our isolates with to do that source tracking. Additionally, we have supplied the data that could possibly do what you are asking. The WGS data for all of these isolates have been released to public databases at NCBI. Anyone can access those sequences and do whole genome SNP typing which may have the resolution that MLST lacks that would be able to do the kind of study you suggest. This is demonstrated in Figure 3. Of course, we still run into the problem that there are very little WGS data to compare to from this region. So far that is; the future may be very different! Therefore, we believe this is beyond the scope of the current study, but we hope to continue this work and may be able to do this in a few years or maybe a decade.

Round 2

Reviewer 1 Report

I have no further comments

Reviewer 2 Report

The authors have provided revisions that are sufficient for publication. Thank you for your cooperation.